# Archival influenza virus genomes from Europe reveal genomic variability during the 1918 pandemic

Livia V. Patrono[1,2,19], Bram Vrancken [3,19], Matthias Budt[4,19], Ariane Düx[1,2], Sebastian Lequime [5], Sengül Boral[6], M. Thomas P. Gilbert [7,8], Jan F. Gogarten[1,2], Luisa Hoffmann[4], David Horst[6], Kevin Merkel[1,2], David Morens[9], Baptiste Prepoint [2,10], Jasmin Schlotterbeck[2], Verena J. Schuenemann[11], Marc A. Suchard [12,13,14], Jeffery K. Taubenberger [15], Luisa Tenkhoff[4], Christian Urban[11], Navena Widulin[16], Eduard Winter[17], Michael Worobey [18], Thomas Schnalke [16,20], Thorsten Wolff [4,20], Philippe Lemey [3,20] & Sébastien Calvignac-Spencer [1,2,20 ✉]

The 1918 influenza pandemic was the deadliest respiratory pandemic of the 20th century and determined the genomic make-up of subsequent human influenza A viruses (IAV). Here, we analyze both the first 1918 IAV genomes from Europe and the first from samples prior to the autumn peak. 1918 IAV genomic diversity is consistent with a combination of local transmission and long-distance dispersal events. Comparison of genomes before and during the pandemic peak shows variation at two sites in the nucleoprotein gene associated with resistance to host antiviral response, pointing at a possible adaptation of 1918 IAV to humans. Finally, local molecular clock modeling suggests a pure pandemic descent of seasonal H1N1 IAV as an alternative to the hypothesis of origination through an intrasubtype reassortment.

[1] Epidemiology of Highly Pathogenic Microorganisms, Robert Koch Institute, Berlin, Germany. [2] Viral Evolution, Robert Koch Institute, Berlin, Germany. [3] Laboratory of Clinical and Epidemiological Virology, Department of Microbiology, Immunology and Transplantation, Rega Institute, KU Leuven, Leuven, Belgium. [4] Unit 17 Influenza and other Respiratory Viruses, Robert Koch Institute, Berlin, Germany. [5] Cluster of Microbial Ecology, Groningen Institute for Evolutionary Life Sciences, University of Groningen, Groningen, The Netherlands. [6] Institute for Pathology, Charité, Berlin, Germany. [7] Center for Evolutionary Hologenomics, The GLOBE Institute, University of Copenhagen, Copenhagen, Denmark. [8] University Museum, NTNU, Trondheim, Norway. [9] Office of the Director, National Institute of Allergy and Infectious Diseases, Bethesda, MD, USA. [10] Département de Biologie, Ecole Normale Supérieure, PSL Université Paris, Paris, France. [11] Institute of Evolutionary Medicine, University of Zurich, Zurich, Switzerland. [12] Department of Biostatistics, Fielding School of Public Health, University of California, Los Angeles, Los Angeles, CA, USA. [13] Department of Biomathematics, David Geffen School of Medicine, University of California, Los Angeles, Los Angeles, CA, USA. [14] Department of Human Genetics, David Geffen School of Medicine, University of California, Los Angeles, Los Angeles, CA, USA. [15] Viral Pathogenesis and Evolution Section, Laboratory of Infectious Diseases, National Institute of Allergy and Infectious Diseases, Bethesda, MD, USA. [16] Berlin Museum of Medical History, Charité, Berlin, Germany. [17] Pathological-anatomical collection in the Narrenturm, Natural History Museum of Vienna, Vienna, Austria. [18] Department of Ecology and Evolutionary Biology, University of Arizona, Tucson, AZ, USA. [19]These authors contributed equally: Livia V. Patrono, Bram Vrancken, Matthias Budt. [20]These authors jointly supervised this work: Thomas Schnalke, Thorsten Wolff, Philippe Lemey, Sébastien Calvignac-Spencer. ✉email: calvignacs@rki.de

The 1918 influenza A (H1N1) pandemic (hereafter 1918 pandemic) was the largest global catastrophe of viral origin in the last century. Medical and historical reports from that period have long represented the only direct source of information on this event, which caused the death of 50–100 million people worldwide[1]. Based on these, we know that the disease was recognized nearly simultaneously across continents in the summer of 1918, peaked in the autumn of 1918 and continued through the winter of 1919. While young children and the elderly were severely affected, the 1918 pandemic stood out as causing exceptionally high mortality in healthy 20–40 years-old people. Its duration, high death toll and unusual epidemiology all contributed to its profound impact on societies of that time[2].

That the pandemic was caused by a virus was already speculated in 1918[3], but only finally proven in the 1930s, when evidence of disease associated with influenza viruses was obtained. The characterization of viruses from the 1918 pandemic only began much later once suitable molecular analysis tools were developed. In the late 1990s, studies on formalin-fixed, paraffin-embedded tissue blocks and permafrost-preserved bodies from victims of the 1918 pandemic ascertained that the causative agent was an influenza A virus (IAV) of the H1N1 subtype[4,5]. Since then, two complete IAV genomes have been reconstructed from victims who died in September 1918 at Camp Upton, New York (hereafter CU)[6] and in November 1918 at Brevig Mission, Alaska (hereafter BM)[7]. In addition, 15 partial and one complete sequence of the hemagglutinin (HA) gene have been obtained from other influenza victims who succumbed between May 1918 and February 1919 in the USA ($n = 14$) and the UK ($n = 2$)[8,9].

These sequences offered some insights into the origins and aftermaths of the pandemic, as well as into the virus phenotype. At least seven segments (all except HA) harbored by the 1918 virus appeared to be drawn from the diversity of IAV strains circulating in an avian reservoir and these were passed on to seasonal H1N1 influenza viruses[10,11]. The reconstruction of an infectious 1918 virus based on the BM genome sequence showed that the HA and polymerase complex genes were likely major determinants of this virus pathogenicity[12,13]. Yet, many questions remain unaddressed or are still debated, largely as a result of the paucity of sequence information. For example, how much genomic diversity was generated during the pandemic? Was the change in severity observed between the pre- and pandemic peak periods ascribable to distinct viral genetic features? And, what was the origin of HA in subsequent seasonal H1N1 viruses?

Obtaining a large collection of genomes is likely beyond reach since permafrost-preserved bodies and pathology specimens of victims of the pandemic are rare. While the few sequences we might be in the position to generate may well be insufficient to comprehensively answer the aforementioned questions, they may still offer new valuable insights if representative of key timepoints and locations in the epidemic. Therefore, we applied recent advances in nucleic acid recovery from archival samples[6,14] to as-yet unexplored European pathology collection material, including the medical archive started by Rudolf Virchow in mid 1800s Germany[15]. This enabled us to sequence two partial and one complete 1918 influenza virus genomes from specimens sampled in Berlin and Munich, two of which derive from the pre-pandemic peak period, which we used to start revisiting a number of hypotheses about the 1918 influenza virus.

## Results and discussion

**1918 influenza virus genomes from Munich and Berlin.** To explore the emergence, pandemic and post-pandemic periods, we selected 13 formalin-fixed lung specimens dated between 1900 and 1931, preserved within the Berlin Museum of Medical History at the Charité (Berlin, Germany, $n = 11$) and the pathology collection (Narrenturm) of the Natural History Museum in Vienna, Austria ($n = 2$). This set included six specimens collected during pandemic years in Europe ($n = 4$ in 1918, $n = 2$ in 1919). Details about all specimens, including initial diagnosis, are presented in Table 1 and in the Methods section. For each specimen, we heat treated 200 mg of formalin-fixed lung tissue to reverse macromolecule cross-links induced by formalin[16], then performed nucleic acid extraction. Following DNase treatment and ribosomal RNA depletion, we built high-throughput sequencing libraries and shotgun sequenced them on Illumina® platforms.

We identified IAV reads in libraries from 3 of 13 specimens, all of which date to 1918 and were associated with histopathological findings of bronchopneumonia (Fig. 1a; Supplementary Note 1 and Supplementary Fig. 1). Viral RNA preservation was generally good, as shown by median fragment lengths well above 100 nucleotides and the lack of any obvious damage pattern (Supplementary Figs. 2–3). Human RNA fragments were on

**Table 1 Details on the archival samples used in this study.**

| Sample ID | Date | Location | Collection | Pathological diagnosis | 1918 IAV detection |
|---|---|---|---|---|---|
| 18.560/684 | 06.01.1900 | unknown | Narrenturm, Vienna | Influenza pneumonia | Negative |
| 447 | 1913 | unknown | Charité, Berlin | Confluent pneumonia | Negative |
| 876 | 1913 | unknown | Charité, Berlin | Calluses of the pleura | Negative |
| 1150 | 1914 | unknown | Charité, Berlin | Tuberculosis | Negative |
| 928 | 1915 | unknown | Charité, Berlin | Tuberculosis | Negative |
| 162 | 1918 | Munich | Charité, Berlin | Influenza bronchopneumonia | Positive<br>Complete genome at 1944x average coverage depth |
| 572 | 27.06.1918 | Berlin | Charité, Berlin | Tracheobronchitis, purulent hemorrhagic bronchopneumonia | Positive<br>89.3% of the genome at 53x average coverage depth |
| 576 | 27.06.1918 | Berlin | Charité, Berlin | Hemorrhagic bronchopneumonia, fibrinous bronchitis | Positive<br>57.2% of the genome at 9x average coverage depth |
| 1068 | 1918 | unknown | Charité, Berlin | Fibrinous pseudomembranous bronchitis | Negative |
| 84 | 1919 | unknown | Charité, Berlin | Caseous pneumonia | Negative |
| 247 | 1919 | Munich | Charité, Berlin | Tuberculosis | Negative |
| 112 | 1920 | unknown | Charité, Berlin | Influenza bronchopneumonia | Negative |
| 15.929 | 1931 | unknown | Narrenturm, Vienna | Influenza pneumonia | Negative |

a

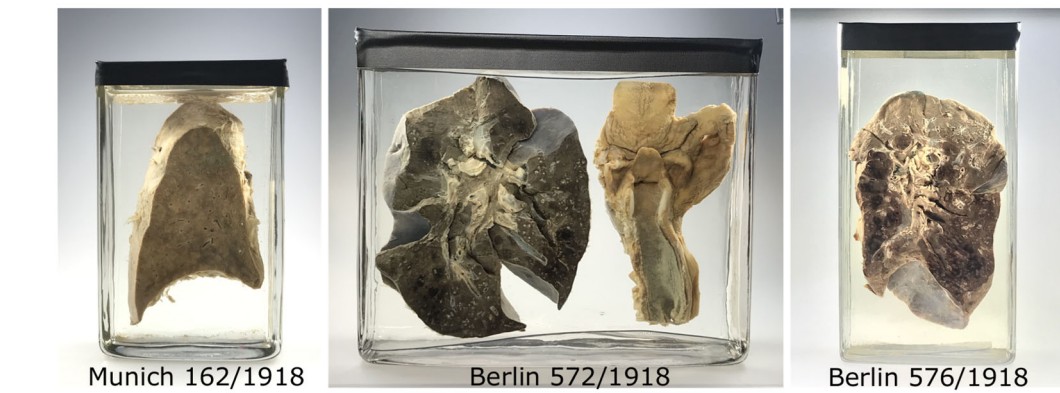

Munich 162/1918 Berlin 572/1918 Berlin 576/1918

b

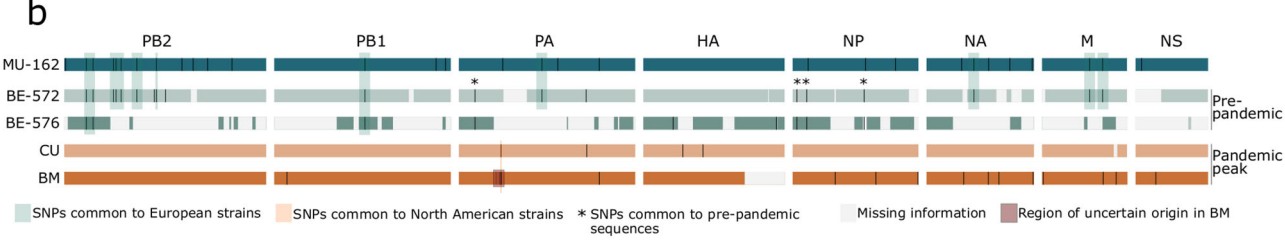

| | SNPs common to European strains | | SNPs common to North American strains | | * SNPs common to pre-pandemic sequences | | Missing information | | Region of uncertain origin in BM |

**Fig. 1 Positive specimens and influenza A virus genomic coverage.** Specimens positive for influenza A virus (**a**) and comparison of all available 1918 influenza A virus genomes (**b**). We identified single nucleotide polymorphisms (SNPs) in the new genomes using BM as a reference; for BM and CU, we plotted SNPs unique to these genomes. Missing information represents areas where we did not get any coverage or this was lower than our consensus calling criteria.

average shorter, possibly indicating better protection of viral RNA inside virions (Supplementary Note 2). We did not detect any other viral agent likely to be involved in severe respiratory disease, but identified a number of bacteria which have been previously associated with respiratory diseases (e.g. *Mycobacterium kansasii* and *Pseudomonas aeruginosa* in specimens that did not produce influenza reads) and with 1918 influenza (*Klebsiella pneumoniae, Pasteurella multocida, Staphylococcus aureus* and *Streptococcus pneumoniae*); in 2 specimens bacterial colonization was also evident from histopathology (Supplementary Note 1 and Supplementary Fig. 1).

Increased sequencing efforts for the three influenza-positive samples MU-162 (Munich), BE-572 and BE-576 (Berlin) yielded 40.133.161, 31.989.479 and 14.965.377 high quality reads, respectively. Of these, 55.5%, 0.33% and 0.1% mapped to the influenza A/Brevig Mission/1/1918 virus, which allowed us to reconstruct a complete 1918 influenza virus genome at a 1944x average coverage depth for MU-162, as well as significant portions of the genomes for BE-572 and BE-576 (89.3% and 57.2%, at an average coverage depth of 53x and 9x, respectively).

**Genomic variation of 1918 viruses and potential impact on polymerase complex activity.** Together with the available BM and CU sequences, we first used these German influenza genomes to assess the genomic diversity of strains sampled in different geographical locations and months. A genomic map summarizing all nucleotide differences is presented in Fig. 1b (detailed discussion in Supplementary Note 3). The two partial genomes from Berlin sampled in June 1918 differed at a maximum of two sites, in HA (Supplementary Fig. 4), suggesting that within-outbreak variability may have been negligible at this location and timepoint. We acknowledge that this comparison was based on only a third of the viral genome, potentially representing an

underestimation of local diversity. When comparing the nearly complete BE-572 genome to the other German sequence from Munich (MU-162), 22 SNPs were identified, resembling the difference previously reported for the two North American genomes[6]. Comparison between European and North American genomes revealed 22-43 SNPs. Finally, when comparing the pre-pandemic peak (June 1918) European BE-572 genome with pandemic peak genomes CU (September 1918) and BM (November 1918), we identified 22 and 29 SNPs, respectively. Overall, these comparisons show that measurable genomic variability was present, and that genomes sampled on the same continents (0.11-0.16%) and during the same period of the pandemic (0.11%) exhibited lower overall divergence than genomes sampled on different continents (0.16-0.32%), and during different periods (0.16-0.21%). This pattern is compatible with spatio-temporal effects of local transmission.

To evaluate whether the genomic differences observed may have had a phenotypic impact, we conducted exploratory investigations in vitro. To the best of our knowledge, no such comparisons have been made between the genomes from North America. We focused on the MU-162 virus, for which we obtained a high-quality complete genome sequence, and on the BM virus, the only 1918 IAV ever reconstructed and characterized phenotypically[12]. Since the polymerase complex contained most of the coding mutations found in MU-162 (4 in PB2, 2 in PB1, 3 in PA and 1 in NP proteins, Supplementary Fig. 4), and is a major pathogenicity factor for BM, we resynthesized viral genes and performed a functional comparison of the activity of BM and MU-162 polymerases after reconstitution in transfected cells. Using reporter assays, dose response curves revealed a two-fold higher activity of BM compared to MU-162 polymerase (Fig. 2a).

To identify the subunit(s) responsible for this difference, we determined the effect of swapping single polymerase subunits between BM and MU-162 (Fig. 2b). The introduction of PA from

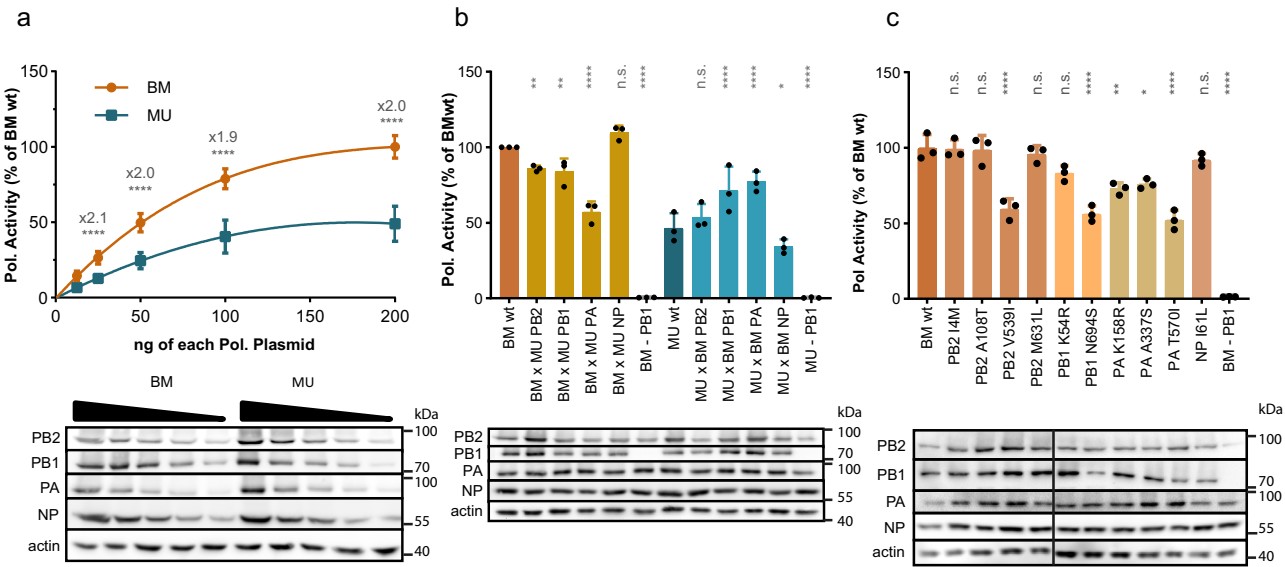

**Fig. 2 In vitro activity of reconstituted BM and MU-162 polymerases by mini-genome reporter assay. a** Dose response of transfected MU-162 (here referred to as MU) polymerase in comparison to BM activity, arbitrarily set to 100%. **b** Effect of whole segment swapping. Comparative analysis of BM and MU-162 polymerase complexes expressed as wild type (wt) or as reassorted 3 + 1 combinations. **c** Effect of single BM-to-MU-162 aa changes in the context of the BM polymerase. Each assay was conducted in $n = 3$ independent replicates with experimental triplicates. Values are means ± SD. Significance was determined by nested ANOVA. *$P < 0.05$; **$P < 0.01$; ****$P < 0.0001$. All three lower panels depict the expression levels of polymerase subunits by immunoblot analyses. A representative result of $n = 3$ independent assays is shown. Source data are provided as a Source Data file.

MU-162 into the BM polymerase complex caused a 1.7-fold reduction of its activity (Fig. 2b, $P < 0.0001$). In line with this finding, BM PA increased the activity of the MU-162 polymerase by 1.7-fold ($P < 0.0001$). Exchange of PB1 and PB2 subunits caused a similar, but more modest impact on activity. No effect was recorded upon exchange of NP proteins.

To more precisely pinpoint the genetic determinants of the observed phenotypic differences, we generated BM point mutants, each carrying one of the amino acid (aa) changes identified in the MU-162 virus. Five mutations distributed across PA (3), PB1 (1) and PB2 (1) significantly reduced the activity of the polymerase compared to BMwt (Fig. 2c). These results indicate that, in vitro, amino acid changes differentiating MU-162 and BM affect the activity of the viral RNA polymerase complex, which has a critical role in balancing viral virulence[17].

Assessing whether the observed differences would translate into distinct phenotypes comes with a lot of uncertainty. Similar fold differences observed for other highly pathogenic IAVs isolated from humans have been associated with an increase in virus replication and virulence[18]. Yet, it is also known that polymerase activity and viral virulence are not always linearly correlated[17,19]. In addition, our analyses also pinpointed the uncertain origin of a mutation found in BM PA (Supplementary Note 3). With these caveats in mind, we conclude that if our findings do not necessarily predict functional differences in vivo, they are, at this stage, compatible with the notion of phenotypically polymorphic 1918 influenza viruses and should encourage future research on the question, using more viral sequences and possibly reconstructed viruses.

**Geographic spread and functional evolution of the 1918 virus during the pandemic.** To investigate to what extent the collective evidence generated thus far informs us about how the pandemic unfolded, we combined the newly generated genomes with previously published 1918 sequences. We first focused on the question of the geographic spread of the 1918 virus. For this, we took advantage of the availability of a total of 21 HA sequences

sampled in Europe and North America before and during the pandemic period to infer a time-scaled evolutionary history in a Bayesian phylogenetic framework[20]. This reconstruction showed the interspersed clustering of the three German sequences with the North American ones, suggesting no geographic segregation between continents (Fig. 3). Similarly, two sequences sampled three months apart in London during the 1918-1919 pandemic winter (November 1918 and February 1919, respectively) clustered separately, albeit with moderate support (posterior probability 0.88, Fig. 3 and Supplementary Fig. 5). Models in which monophyletic clustering was enforced on the European lineages showed a decisively lower model fit (ln Bayes factor [BF] ≥ 8.9, Supplementary Note 4), supporting the hypothesis of transatlantic movement of pandemic viruses, in line with the historical context of human movement near the end of World War I. Altogether, genomic data support a scenario of predominantly local transmission with frequent long-distance dispersal events.

Next, we focused on the evolutionary relationships between viruses of the different periods of the 1918 pandemic. For this, we used the same HA sequence dataset, this time to assess whether pandemic peak viruses descended from a single global replacement of pre-pandemic peak viruses. Under such a scenario, the pandemic peak viruses should constitute a single monophyletic lineage. However, the data strongly support the alternative scenario (ln BF support ≥ 10.1), where multiple pre-pandemic peak lineages survived into the following pandemic months. Both the lack of strong geographical stratification, and the survival of initial lineages over a time frame covering pre- and post-pandemic peak periods (280 days) are dynamics shared with the 2009 H1N1 pandemic (Supplementary Note 4).

We then specifically searched for potential adaptations of the virus that could have arisen between the pre- and pandemic peak period and might have contributed to the increased severity of the latter[2]. The presence of the aa residue G222 in the receptor binding domain of the H1 subtype HA protein has already been discussed in this context[8]. This residue confers a binding affinity for both avian and human glycans, while the human-like D222 only efficiently binds human glycans[21,22]. In a recent study, three

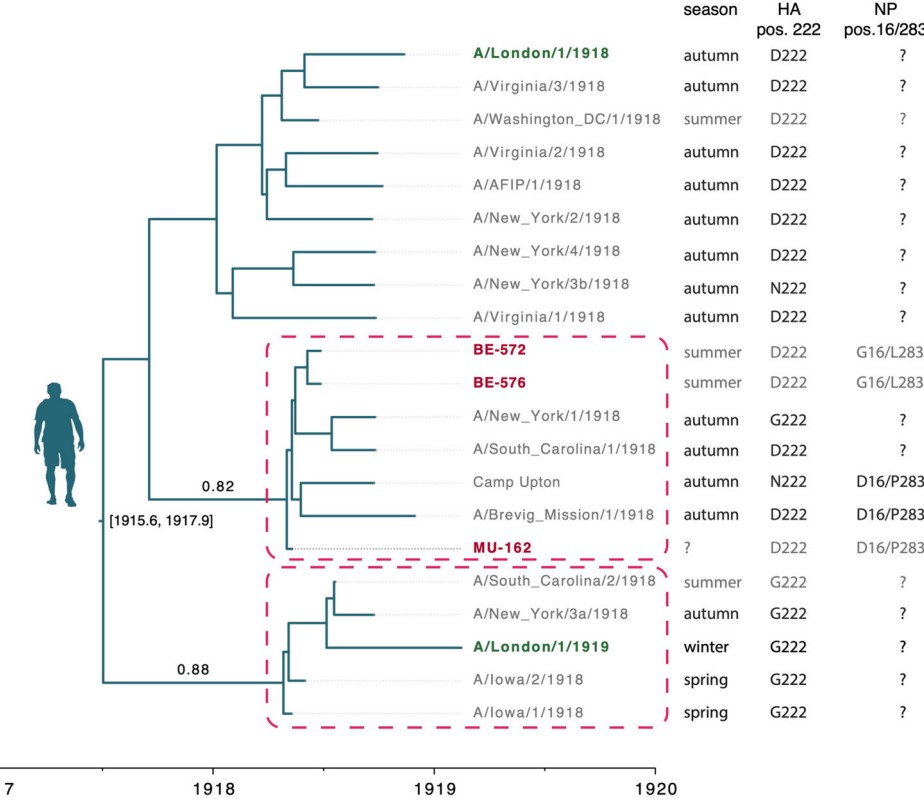

**Fig. 3 Time-scaled phylogeny reconstructed based on HA sequences from 1918 flu strains.** US sequences are in gray, European sequences are in dark red (Germany) and dark green (UK). Dashed rectangles highlight clades comprising strains from different continents and with posterior support ≥0.75; for these clades posterior probabilities are reported above stem branches. The season column indicates from which season the samples originate, with light gray for the pre-pandemic peak period (spring and summer) and dark gray for the pandemic peak period (autumn and winter). The amino acid residues at HA position 222 and NP positions 16 and 283 are also indicated. '?' indicate the absence of information. Numbers between brackets next to the root node indicate the 95% Highest Posterior Density (HPD) of its estimated age. The BE-576 consensus genome with ambiguities was used in this analysis; an equivalent reconstruction using majority rule consensus base calling can be found in Supplementary Fig. 5.

of four pre-pandemic peak sequences presented G222 in contrast to only two of nine pandemic peak sequences (all other sequences presented D222)[8]. The three German HA sequenced here carried the human-like D222, which reduces the likelihood that the frequency of the two variants varied significantly between the pre- and pandemic peak periods.

The availability of genome-wide information about two 1918 summer strains allowed us to identify two other aa changes that potentially mark an important difference between pre- and pandemic peak periods. We found that pre-peak strains BE-572 and BE-576 carried avian-like residues at position 16 (G) and 283 (L) of NP, whereas the peak strains and MU-162 (unknown sampling date) carried D16 and P283, as found in most human H1N1 strains with the exception of the 2009 pandemic H1N1 virus (see references[23–26] and Supplementary Fig. 6). These two sites are known to influence host range and susceptibility/resistance to the interferon-induced MxA antiviral protein[23,24,27]: position 283 is located in the body of NP, and a proline at this site confers resistance to the human MxA protein, to which D16 also contributes, albeit to a lower extent[24,26]. These mutations may represent hallmarks of early adaptation to humans: during the first months of their spread, 1918 influenza viruses may have evolved a better capacity to evade the innate interferon response, which is an important aspect of influenza virus pathogenicity.

**1918 influenza viruses and the origin of human seasonal H1N1 influenza viruses.** The sequencing of BM offered the opportunity to test for a pandemic origin of seasonal H1N1 segments. It is to

be expected that in phylogenetic trees the pandemic origin of a segment should translate into BM being basal to seasonal influenza, and that in time-scaled phylogenetic trees their most recent common ancestor (MRCA) should then date back to or shortly precede pandemic years. Multiple modeling approaches have been implemented to investigate the question and their output have been interpreted as supporting different scenarios[5,11]. The latest of these efforts made use of molecular clock models allowing for host-specific rates of nucleotide evolution (thereafter host-specific local clocks; HSLC). These suggested that seven seasonal H1N1 segments originated in birds, while the eighth (HA) arose from a co-circulating homosubtypic H1 IAV through intrasubtype reassortment, in agreement with other non-genetic information[11].

We revisited this hypothesis with our extended sequence data set, that now comprised multiple 1918 virus sequences for all segments, using approaches incorporating or not a molecular clock model. Non-clock maximum likelihood (ML) reconstructions indicated that human seasonal H1N1 and 1918 pandemic viruses cluster together with reasonably high bootstrap support, with the seasonal lineage nested within the 1918 pandemic variants for all segments but PB2 (Supplementary Fig. 7). Time-measured phylogenetic inference using a HSLC model found that for all internal gene segments the human seasonal lineage nests within pandemic flu diversity, while for HA and NA pandemic viruses form a monophyletic cluster with swine influenza viruses, which is positioned as a sister lineage to human seasonal lineage[11] (Fig. 4a vs b, Supplementary Fig. 8).

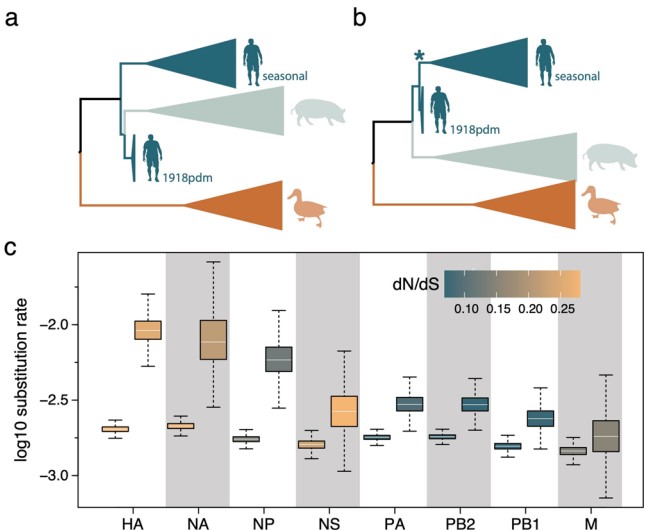

**Fig. 4 Time-measured phylogenetic patterns and evolutionary rates estimated using Bayesian molecular clock modeling. a** The phylogenetic pattern for HA and NA inferred using a standard host-specific local clock (HSLC) model. **b** A monophyletic cluster for human pandemic and seasonal viruses estimated for HA and NA using an extended HSLC model (and using either model for all other segments). The star denotes the branch that is allowed to have a separate evolutionary rate in the HSLCext model. **c** Evolutionary rate estimates for the human lineage (first box) and the seasonal ancestral branch (second box) under the extended HSLC model for each segment ordered according to difference in these two rates. The boxes are colored according to the dN/dS estimate for the human lineage and the seasonal ancestral branch in each segment. The horizontal line in the whisker plots represents the mean. The lower and upper bounds of the boxes indicate the first and third quartile, respectively. Vertical lines are the upper and lower whisker representing the minimum of the largest value and 1.5 times the inter quartile distance, respectively the maximum of the smallest value and 1.5 times the inter quartile distance. Sample size from the MCMC chain is 2702 for NA and 4502 for all other segments. Source data are provided as a Source Data file.

Because of the apparent conflict between the HA and NA phylogenies inferred with both approaches (Fig. 4a, Supplementary Fig. 9), we explored an alternative scenario that allows for a higher evolutionary rate on the branch ancestral to the seasonal lineage. Such a scenario would result in considerably higher divergence between pandemic and seasonal viruses for H1 and N1 than would be expected under a strict clock, and could therefore induce a sister lineage pattern with a relatively deep MRCA under the HSLC model - which assumes a different but constant rate of evolution in each of the host-specific lineages. We ran simulations that showed that if the rate is elevated on the relevant branch, and this is not accounted for in the clock model, the HSLC inference generally fails to recover the seasonal lineage as a direct descendant of the pandemic virus (Supplementary Fig. 10). Fitting an extended HSLC model that allows for a separate rate on the relevant branch inferred the seasonal lineage as a descendant of the pandemic virus in both HA and NA, consistent with the non-clock trees.

Across segments, this model resulted in consistently higher posterior mean rates on the seasonal ancestral branch compared to the seasonal clade rates (Fig. 4), but with a variability that largely reflected the variation in selective constraints on the segments. Less stringent purifying selection in HA and NA, as reflected in higher nonsynonymous/synonymous substitution rate ratios (Fig. 4c), accompanied by a stronger rate acceleration potentially explains the observed patterns. To investigate whether

variation in selection pressures could explain the acceleration on the branch leading to the seasonal lineage, we performed selection analyses using codon substitution models. These models did not identify diversifying episodic selection or relaxed selective constraints on this branch in any of the segments, implying that only mutation rate/generation time effects may explain the branch-specific elevated rate. Interestingly, a higher mutation rate for the seasonal predecessor (due to a higher replication rate) has indeed been suggested by some experimental evidence comparing BM to seasonal strains[12,28].

Altogether, these new analyses are compatible with the scenario of a pure pandemic origin of seasonal H1N1 viruses. However, the essentially phenomenological nature of our modeling approach does not, for now, allow us to definitely favor it over the alternative scenario of a homosubtypic reassortment.

In general, we acknowledge that, due to our very small sample size (three complete and two partial genomes), all findings reported here remain preliminary. Additional genomes from archival samples surrounding the pandemic period, as well as phenotypic characterization of multiple 1918 viruses in vitro and in vivo, will undoubtedly provide the opportunity for more robust tests of our hypotheses. The fact that three out of the six archival samples we analyzed from pandemic years yielded good quality viral RNA underscores the enormous potential of genomic research of medical collections, which can now be considered a low risk-high gain enterprise. Therefore, we anticipate that the main obstacle to a better understanding of the evolution of 1918 flu viruses will lie in the identification of preserved pathological specimens. Unfortunately, such specimens are rare and hard to localize, which highlights the importance of long-neglected museum activities[29]. Guaranteeing proper long-term storage conditions of biological specimens, as well as digitizing and transcribing the associated written sources (e.g. dissection protocols), are essential steps to take to preserve and use the remaining sparse collections still holding substantial archival material.

## Methods

**Samples**. Ethics approval was obtained from the ethics committee of the Charité (Berlin, Germany) under the reference number EA4/212/19. We obtained 11 lung samples from formalin-fixed specimens identified in the collection of the Berlin Museum of Medical History at the Charité (Berlin, Germany). Sample 447/1913 (sample ID/year of collection) was taken from a 21-year-old (yo) male with a pathological report of fresh confluent pneumonia of the lower lung lobes. Sample 876/1913 was taken from a 45-yo male with a pathological report of calluses on the pleura. Samples 1150/1914 and 928/1915 were taken from a 16-yo and 9½ month-old male, respectively, with a pathological report of tuberculosis. Sample MU-162 was taken from a 17-yo female who died of influenza-related pneumonia in Munich in 1918. The exact collection date was not reported. The pathological finding reported the presence of purulent bronchitis, bronchiolitis and bi-lateral confluent bronchopneumonia. Sample BE-572 was taken from a 18-yo male soldier who died in Berlin on June 27th 1918. The pathological findings were severe purulent pseudomembranous tracheobronchitis and purulent hemorrhagic bronchopneumonia. Sample BE-576 was taken from a 17-yo male soldier who died of influenza in Berlin on June 27th 1918. The pathological findings were fibrinous bronchitis and purulent hemorrhagic bronchopneumonia. Sample 1068/1918 was taken from a 13-yo female. The main pathological finding was fibrinous pseudo-membranous bronchitis. Sample 84/1919 was taken from a 55-yo male who died with a caseous pneumonia of the left pulmonary lobe. Sample 247/1919 was taken from a 32-yo woman with tuberculosis who died in Munich. Pathological findings included chronic lung tuberculosis with bronchiectasis, typical lung cavernae and pneumonia. Sample 112/1920 was taken from a 6-month-old individual (gender not reported) who died of influenza. The pathological report included purulent bronchitis and bronchopneumonia, double-sided empyema and collapse of both lungs. Since the exact composition and concentration of the formalin fixative was unknown, we stored the lung samples in PBS to avoid further damage to nucleic acids by adding fresh formalin. Our sample set was further expanded with two formalin-fixed lung specimens preserved within the pathology collection (Narrenturm) of the Natural History Museum in Vienna, Austria, cataloged as influenza-related pneumonia. Sample 18.560/684 originated from the Spital Rudolfstiftung in Vienna and was taken from a child who died on January 6th 1900 with a diagnosis of influenza-related pneumonia. Sample 15.929 originated from

the University of Graz and was taken from a 23-yo man who died in 1931. The pathological findings were confluent influenza-related pneumonia with gangrenous cavernae and acute empyema. After sampling, the specimens from the Narrenturm were preserved dry.

**Sample preparation**. To minimize contamination, all steps prior to sequencing were performed in an ancient DNA laboratory.

*RNA extraction*. To maximize chances of viral RNA recovery, we performed 8 separate total nucleic acid extractions from different areas of the lung using the DNeasy® Blood & Tissue Kit (Qiagen) with modifications for formalin-fixed samples, that has been demonstrated to effectively recover both RNA and DNA from such samples[30]. For each separate extraction, a pea-sized piece (ca. 25 mg) of lung was washed in 1 ml PBS to remove residual fixative. The washed tissue was cut into smaller pieces using sterile scissors and added to bead tubes containing tissue lysis buffer (ATL). To reverse formalin-induced crosslinking[16], the tissue was heated to 98 °C for 15 min. To facilitate lysis, the tissue was homogenized by bead beating with a Fast Prep® (MP Biomedicals). We added 20 µl Proteinase K and kept the homogenate at 56 °C until the tissue was completely lysed (ca. 1 h). Subsequent steps were performed according to protocol and nucleic acid was eluted in 35 µl elution buffer (AE).

*Library preparation*. To maximize viral RNA in the final sequencing libraries, we removed DNA and ribosomal RNA from the nucleic acid extracts before conversion to double-stranded cDNA. For DNase treatment we used the TURBO DNA-free™ Kit (Ambion). To reduce costs for rRNA depletion, 4 DNase-treated extracts were pooled and concentrated using RNA Clean & Concentrator-5 Kit (Zymo Research) and eluted in 13 µl nuclease free water as input for one ribosomal RNA depletion reaction. The other 4 extracts were treated separately. We performed ribosomal RNA depletion and clean-up using the NEBNext® rRNA Depletion Kit (Human/Mouse/Rat) with RNA Sample Purification Beads (New England Biolabs) according to protocol. Following bead clean-up RNA was eluted in 12 µl nuclease free water. We performed cDNA synthesis, using the Super-Script™ IV First-Strand Synthesis System (Invitrogen) and converted cDNA into ds DNA with the NEBNEXT® mRNA Second Strand Synthesis Module (New England Biolabs). Double-stranded DNA was purified using MagSi-NGS^prep Plus Beads (Steinbrenner Laborsysteme) and eluted in 50 µl TET. We prepared 5 separate libraries with the NEBNext® Ultra™ II DNA Library Prep Kit for Illumina® (New England Biolabs) without prior fragmentation of double-stranded cDNA and without size-selection upon adapter ligation. All clean-up steps during the library preparation were conducted with MagSi-NGS^prep Plus Beads (Steinbrenner Laborsysteme). The libraries were dual indexed with NEBNext® Multiplex Oligos for Illumina® (New England Biolabs), quantified using the KAPA Library Quantification Illumina Universal Kit (Roche), amplified with the KAPA HiFi HotStart ReadyMix (Roche) and Illumina adapter-specific primers, and diluted to a concentration of 4 nM for sequencing.

*Sequencing*. Libraries were sequenced on an Illumina® MiSeq platform using the v3 chemistry (2 × 300-cycle) and on the Illumina® NextSeq platform using v2 chemistry (2 × 150-cycle).

**NGS data analyses**. Raw reads were pre-processed using Trimmomatic[31], with the following settings: LEADING:30 TRAILING:30 SLIDINGWINDOW:4:40 MIN-LEN:40. Reads were initially classified using Kraken2[32]. Trimmed reads were then mapped to the A/Brevig Mission/1/1918 (BM) reference genome (GenBank accession numbers AY130766, AF333238, AF250356, AF116575, AY744935, DQ208309-11) using BWA-MEM[33]. Upon detection of influenza reads in sample MU-162, we proceeded with de novo assembly using metaSPAdes[34]. Scaffolds greater than 750 bp (the smallest BM segment is 830 bp) were blasted against a database of influenza virus sequences and all 8 viral segments were identified. The 8 segments resulting from the de novo assembly were then used as a reference for reference-based mapping of reads generated from sample MU-162 and all other samples included in this study. To minimize the effects of cross-library sequence bleed-through, we discarded data where samples were co-sequenced with MU-162. Upon trimming, paired end reads were merged using the Clip&Merge tools from EAGER[35]. We sorted mapping files and removed duplicates using the SortSam and MarkDuplicates tools from Picard (http://broadinstitute.github.io/picard). We assessed RNA damage using mapDamage v2[36]. Base calling for MU-162 was set to 20 reads and 95% agreement whereas for BE-572 and BE-576 to 3 reads and 50% agreement. Recombination analyses for the PA segment were run in Rdp4 using the PA dataset assembled by Worobey et al.[10] We reduced the dataset to unique sequences using FaBox[37] and then chose the 99 most ancient ones to analyze with the MU-162 sequence. We ran Rdp4[38] with all methods (changing the settings of the bootscan approach to a window size of 100). Geneconv, bootscan, lard and 3seq identified Brevig as a recombinant (MU-162 backbone with a ca. 100 bp originating in another virus). PhyML[39] was finally run on the backbone and recombinant region.

The influenza positive samples were analyzed for human RNA preservation. The sequencing data was processed using EAGER[35]. Sequencing reads were

inspected with FastQC before merging and adapter trimming with AdapterRemoval V2.2.1a. Pre-processed reads were then aligned to the human transcriptome reference (Gencode, Release 34), using BWA with a minimum quality score of 0 and a maximum edit distance of 0.01. The MarkDuplicates method was used for duplicate processing (Broad Institute) as well as DamageProfiler v0.3.8 and Qualimap[40] to investigate the read length distribution and the damage patterns.

In agreement with the ethics committee of the Charité, Berlin, human reads were removed from the sequencing files prior to uploading them to the European Nucleotide Archive. Filtered reads were mapped to the human RefSeq genome assembly GRCh38.p13 (GCF_000001405.39) using BWA-MEM[33]. Unmapped reads were extracted using SAMtools[41] v1.3.1 and then extracted by name from the original fastq files using seqtk (https://github.com/lh3/seqtk).

**Evolutionary analyses and simulations**
*Datasets*. For the HA and NA segments, the newly generated pandemic H1N1 sequence data were complemented with the collection of human, swine and avian H1 and N1 sequences previously used by Worobey et al.[11]. The A/London/1/1919 isolate[9] was also added to the HA data. The newly sequenced samples from Berlin differ from each other in HA at two positions (alignment positions 356 and 1600). BE-576 is polymorphic at these positions with ~40% of the reads exactly matching the BE-572 HA sequence. It can hence not be excluded that BE-576 is identical to BE-572, and that the majority rule consensus base calls at these positions reflect the amplification of potential RNA damages/RT or PCR errors. For this reason, all HA analyses were run in duplicate, once with the majority rule consensus sequence, and once with these positions represented by the appropriate ambiguity code. For the other segments, except for NS, the new 1918 pandemic flu data were complemented with the human H1N1 viruses, the classical swine flu strains and Eastern plus Western avian lineages from the final datasets used by Worobey et al.[10] downloaded from Dryad. For NS, all human and swine lineages selected by Worobey et al.[10] were used, but only a subset of the avian taxa was taken along to not have to enforce monophyly constraints on the human seasonal and pandemic lineages when accounting for the host-specific evolutionary rates (see below). The region in PA identified as of uncertain origin in the Brevig Mission sequence was not included (masked) for analysis.

*Evolutionary analyses*. Sequences were aligned with MAFFT v7.313[42] and Aliview v1.19[43] was used for manual refinement of the alignments. Maximum likelihood (ML) trees for all datasets were reconstructed using IQTREE 2.0[44] with a general time-reversible (GTR) nucleotide substitution model[45] and a discretized gamma distribution to model among-site rate heterogeneity. Bootstrap support was estimated using the ultra-fast bootstrap procedure with 1000 pseudo-replicates[46]. Time-measured phylogenies were estimated using BEAST v1.10[20] with the same substitution model settings as for the ML trees. The skygrid model was used as a flexible demographic tree prior[47]. To accommodate host-specific rates in the estimation of divergence times, we employed a molecular clock model that allows for different evolutionary rates in lineages of different host species[10]. The standard host-specific local clock (HSLCstd) model specifies the avian influenza virus rate as the background rate, and assumes a different rate for the human (and swine) lineage, and yet another separate rate for the swine lineage specifically. To keep the model identifiable, we constrain a monophyletic cluster for the human and swine lineage as well as a monophyletic cluster for the swine lineage specifically within this cluster. For this specification it remains unclear which rate to assign to the branch ancestral to the human (and swine) lineage (avian or human) and to the branch ancestral to swine lineage (human or swine). Therefore, we integrate over both alternatives for each branch using a stochastic variable selection procedure[48]. To reconcile the time-measured phylogenetic reconstructions with the non-clock ML tree reconstructions for the HA and NA segment, we also extend this model to allow for a separate evolutionary rate along the branch ancestral to the human seasonal lineage. We refer to this model as the extended host-specific local clock (HSLCext). We used BEAGLE 3[49] for efficient likelihood computation and simulate the MCMC chains sufficiently long to ensure stationarity and mixing as diagnosed using Tracer v1.7.0[50]. We summarized posterior tree distributions in the form of maximum clade credibility (MCC) trees and visualize these trees using FigTree (http://tree.bio.ed.ac.uk/software/figtree/). Selection analyses were performed using codon substitution model approaches implemented in HyPhy[51]. Specifically, we tested for episodic (diversifying) selection and relaxed selection along the branch ancestral to human seasonal viruses in the human H1N1 lineage using aBSREL[52] and RELAX[53] respectively. For this purpose, we used the human lineage subtrees extracted from the BEAST MCC trees estimated using the HSLCext model.

*Simulations*. We performed two sets of sequence simulations. First, we simulated sequence data according to the BEAST estimate obtained under the HSLCstd model. We simulated 20 replicate data sets under this scenario (with the same size as the HA data set) and then evaluated the performance of both non-clock maximum likelihood reconstruction and BEAST estimation using the extended HSLC model. Second, we simulated sequence data using the same procedure, but now according to the BEAST estimate obtained under the HSLCext model and evaluated the performance of both non-clock maximum likelihood reconstruction and

BEAST estimation using the standard HSLC model. Simulations were performed using piBUSS[54].

### Functional analyses

*Plasmids.* Complete cDNAs of influenza virus genomic segments derived from sample MU-162 or of strain A/Brevig Mission/1/1918 (H1N1, BM) were commercially synthesized. Point mutations were introduced into BM segments with the Quikchange II Site-directed mutagenesis kit (Agilent, Santa Clara, CA, USA) according to manufacturer's instructions. Coding sequences of polymerase and NP proteins were amplified by PCR with the Phusion Green Hot Start II High Fidelity DNA polymerase (ThermoFisher). PCR products for PB2, PA and NP were cloned into the BsaI sites of vector pCAGGSΔBsa-Blue[55]. PB1, which contains internal BsaI sites, was cloned into the BsmBI sites of a modified version of the pCAGGS plasmid. All constructs were confirmed by Sanger sequencing. Primers used for cloning and mutagenesis are listed in table S1.

*Polymerase activity assay.* Trypsinized 293 T cell cultures were transiently transfected in suspension in 12 wells using lipofectamine 2000 (Invitrogen) with 50 ng of each pCAGGS-pol plasmid, 100 ng pCAGGS-NP plasmid, and 125 ng of pPolI-NS-Luc, expressing an influenza virus-like RNA encoding a firefly luciferase[17]. Transfection was normalized by the constitutively expressed Renilla luciferase encoded on plasmid pTK-RL (10 ng; Promega, Madison, WI, USA). For titration experiments, a range from 12.5 to 200 ng of each pCAGGS-pol construct was used, with respective double amounts of corresponding NP plasmid, while luciferase plasmids were kept constant. Total DNA amount was equalized in every sample with pCAGGS. At 24 h post transfection, luciferase activity was measured with the Dual-Luciferase® Reporter Assays System (Promega, Madison, WI, USA) on a Tristar LB 941 luminometer (Berthold Technologies, Bad Wildbad, Germany) according to manufacturers' instructions.

*Immunoblotting.* Equal amounts of cell lysates from luciferase assay replicates was pooled, denatured in reducing SDS-PAGE sample buffer for 5 min at 95 °C, separated on 8% SDS gels, subjected to immunoblotting with antibodies against PB2 (rabbit, ThermoFisher/Pierce, #PAS-32221), PB1 (rabbit, ThermoFisher/Pierce, #PAS34914), PA (rabbit, GeneTex, #125932), NP (mouse, Acris, #AM00929PU-N) and actin (mouse, Sigma, #A2228), respectively and detected with HRP-labeled anti-mouse (DAKO, #P0260) or anti-rabbit antibodies (DAKO, #P0217), respectively.

*Statistical analysis.* All values are expressed as mean, error bars indicate standard deviations of three independent biological experiments performed in technical triplicates. For multiple comparisons of several groups of a single variable (polymerase subunit exchange in Fig. 2b, mutant analysis in Fig. 2c), nested one-way ANOVA was used and P-values were corrected for multiple testing with Sidak's method. For multiple comparisons of three or more groups with two independent variables (titration of polymerase activity as a function of plasmid concentration in Fig. 2a), nested two-way ANOVA was used and P-values were corrected for multiple testing with Sidak's method. Curve fitting was done with regression analysis using 3rd order polynominal equations. Calculations were performed with GraphPad Prism, version 8.1.2.

**Reporting summary**. Further information on research design is available in the Nature Research Reporting Summary linked to this article.

## Data availability

All raw reads generated for this study have been deposited to the European Nucleotide Archive under project number PRJEB41631 (sample numbers ERS5447402-413, ERS6621911, and ERS5549335-ERS5549361). Human reads were removed as described in the methods section prior to uploading. Kraken2 results for all specimens can be visualized through Krona plots at https://zenodo.org/record/4384755. Nucleotide alignments for the 8 viral segments are available at https://zenodo.org/record/4384715. Source data are provided with this paper.

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

## Acknowledgements

B.V. and S.L. were supported by postdoctoral fellowship grants of the Research Foundation–Flanders (Fonds voor Wetenschappelijk Onderzoek, 12U7121N and 12R4718N, respectively). M.T.P.G. acknowledges Danish National Research Foundation Award DNRF143 for funding. This work was funded in part by the Intramural Research Program of the National Institute of Allergy and Infectious Diseases of the NIH (D.M.M. and J.K.T.). MAS was supported by US National Institutes of Health grants HG006139 and AI135995. M.W. was supported by the Bill and Melinda Gates Foundation (INV-004212) and the David and Lucile Packard Foundation. VJS was supported by the University of Zurich's University Research Priority Program "Evolution in Action: From Genomes to Ecosystems". The research leading to these results has received funding from the European Research Council under the European Union's Horizon 2020 research and innovation program (grant agreement no. 725422-ReservoirDOCS) and from the European Union's Horizon 2020 project MOOD (grant agreement no. 874850). The Artic Network receives funding from the Wellcome Trust through project 206298/Z/17/Z. PL acknowledges support by the Research Foundation - Flanders ('Fonds voor Wetenschappelijk Onderzoek - Vlaanderen', G0D5117N and G0B9317N). This project was also supported by a grant to SCS from the National Research Platform for Zoonoses (Federal Ministry of Education and Research, 01KI1714), and in part by a grant to TW from the Deutsche Forschungsgemeinschaft (Transregio 84, project B2).

## Author contributions

L.V.P., B.V., M.B., T.S., T.W., P.L. and S.C.S. contributed to conceptualization. L.V.P., B.V., M.B., M.T.P.G., M.A.S., M.W., T.W., P.L. and S.C.S. contributed to methodology. L.V.P., B.V., M.B., A.D., J.F.G., L.H., D.H., K.M., B.P., J.S., V.S., L.T., C.U., T.W., P.L. and S.C.S. contributed to the investigation. M.A.S. and P.L. provided software resources. L.V.P., B.V., M.B., S.L., T.S., T.W., P.L. and S.C.S. conducted formal analyses. L.P.V., B.V., M.B., N.W., T.W., P.L. and S.C.S. contributed to visualization. L.V.P., B.V., M.B., D.M., J.K.T., T.S., T.W., P.L. and S.C.S. contributed to validation. N.W., E.W., T.S., T.W., P.L. and S.C.S. provided resources. L.V.P., B.V., M.B., A.D., J.F.G., N.W., T.S., T.W., P.L. and S.C.S. contributed to data curation. L.V.P., B.V., M.B., M.W., T.S., T.W., P.L. and S.C.S. wrote the original draft. All authors contributed to reviewing and editing the manuscript. D.H., V.S., T.S., T.W., P.L. and S.C.S. contributed to supervision. L.V.P. and S.C.S. contributed to project administration. T.S., T.W., P.L. and S.C.S. contributed to funding acquisition. T.S., T.W., P.L. and S.C.S. contributed equally as last authors.

## Funding

## Competing interests

M.A.S. reports contracts from the US Department of Veteran Affairs, the US Food & Drug Administration and Janssen Research & Development, all outside the scope of this work. The remaining authors declare no competing interests.
