## [Peer Review File · Nature Communications]

Reviewer comments, initial review -

Reviewer #1 (Remarks to the Author):

Patrono, Vrancken, Budt, et al. offer a significantly improved manuscript dissecting the early evolution and potential origins of the 1918 influenza pandemic. Their initial submission was an impressive effort to identify and sequence archival samples, yielding the first European isolates of the 1918 pandemic and affording some degree of comparison between early and later waves of the pandemic, and between European and American viral samples. The resubmission provides additional analyses and control experiments that strengthen their original conclusions. Moreover, modifications to the text place the findings in context, highlighting the importance of these discoveries and their limitations. I appreciate the authors' reasoned and thorough response; they have addressed all of my concerns.

Reviewer #2 (Remarks to the Author):

I am really happy to see this manuscript in its revised form. The authors have done an excellent job of addressing my comments. As in my original review, the data are interesting and the methodologies are sound. Now, in this revised manuscript, everything is presented in a much more convincing way with better framing. As with the original, the quality of writing is top notch.

Reviewer #3 (Remarks to the Author):

Patrono et al. provide critical data for understanding the early stages of the 1918 influenza pandemic, its diversification during the pandemic, and subsequent post-pandemic seasonal evolution. The research is timely given the ongoing COVID-19 pandemic. The archival RNA methodology is sound and impressive given the notorious difficulty in retrieving genetic material from formalin-fixed tissues. While I agree with reviewer 2 that the manuscript's conclusions are somewhat limited by the paucity of genetic information available, this manuscript makes a significant contribution from the current status quo of almost no information besides two 1918 H1N1 genomes and some sparsely sampled genic regions. The extension of the analytical clock model is also a critical contribution for future archival virus research.

For the most part, the authors have addressed the previous reviewers' criticisms. I am not qualified to assess the in vitro activity assays, but the authors appear to have addressed the issues identified by both reviewers satisfactorily. The most impressive aspect is that these types of analyses are not typically attempted at all in archival genetic research, so I find it an important step forward for the field beyond in silico hypothesis generation and speculation.

I have two significant outstanding concerns, both raised to some extent by the previous reviewers. First, the phylogenies (and derived models) are determined by only a handful of SNPs. It is therefore critical to show that the results are robust. While I agree that it is valid to leave the hypothesized contaminant region of BM in the analysis, the authors should also consider an Extended Data analysis omitting this region to show that this does not significantly change the results. The authors already did a similar comparison of the tree phylogenies using ambiguities and majority rule consensus and show that the overall results are sound, but the tree topologies do change somewhat.

Secondly, the authors should include the analysis of the data excluding the new genomes for the HSLCstd and HSLCext models and dN/dS values in the supplementary information. I disagree that this is unimportant information -- it raises the critical issue of model misspecification in terms of biological/phylogenetic inference. The outcome of Worobey et al. 2014 changes significantly if the model is incorrect.

Some additional minor suggestions regarding the figures:

Fig 3. Light grey and light blue are difficult to read. Please increase contrast with white background

or change color palette.

Fig 4. I had some difficulty interpreting the meaning of panels A and B at first glance. The figure clarity would be improved by adding a line to the caption explaining that pandemic HA clusters with swine HA in HSLCstd, but with seasonal human HA in HSLCext. Also, the pandemic and seasonal HA use the same symbol. Maybe change the color for the pandemic HA?

I congratulate the authors on their excellent contribution and I look forward to seeing it published in final form.

Michael G. Campana
Smithsonian Conservation Biology Institute

Response to referees' comments

Thank you very much for your comments.

Below you will find our detailed response to your concerns and suggestions. We have numbered all comments and provided annotations in the revised manuscript to facilitate the identification of changes throughout the text.

Reviewer #1 (Remarks to the Author):

1. Patrono, Vrancken, Budt, et al. offer a significantly improved manuscript dissecting the early evolution and potential origins of the 1918 influenza pandemic. Their initial submission was an impressive effort to identify and sequence archival samples, yielding the first European isolates of the 1918 pandemic and affording some degree of comparison between early and later waves of the pandemic, and between European and American viral samples. The resubmission provides additional analyses and control experiments that strengthen their original conclusions. Moreover, modifications to the text place the findings in context, highlighting the importance of these discoveries and their limitations. I appreciate the authors' reasoned and thorough response; they have addressed all of my concerns.

Reply: We are pleased to read that we have properly addressed this reviewer's concerns and thank them for their kind comments.

Reviewer #2 (Remarks to the Author):

2. I am really happy to see this manuscript in its revised form. The authors have done an excellent job of addressing my comments. As in my original review, the data are interesting and the methodologies are sound. Now, in this revised manuscript, everything is presented in a much more convincing way with better framing. As with the original, the quality of writing is top notch.

Reply: We are happy that our revisions are convincing and thank this reviewer for their nice words.

Reviewer #3 (Remarks to the Author):

Patrono et al. provide critical data for understanding the early stages of the 1918 influenza pandemic, its diversification during the pandemic, and subsequent post-pandemic seasonal evolution. The research is timely given the ongoing COVID-19 pandemic. The archival RNA methodology is sound and impressive given the notorious difficulty in retrieving genetic material from formalin-fixed tissues. While I agree with reviewer 2 that the manuscript's conclusions are somewhat limited by the paucity of genetic information available, this manuscript makes a significant contribution from the current status quo of almost no information besides two 1918 H1N1 genomes and some sparsely sampled genic regions. The extension of the analytical clock model is also a critical contribution for future archival virus research.

For the most part, the authors have addressed the previous reviewers' criticisms. I am not qualified to assess the in vitro activity assays, but the authors appear to have addressed the issues identified by both reviewers satisfactorily. The most impressive aspect is that these

types of analyses are not typically attempted at all in archival genetic research, so I find it an important step forward for the field beyond in silico hypothesis generation and speculation.

3. I have two significant outstanding concerns, both raised to some extent by the previous reviewers. First, the phylogenies (and derived models) are determined by only a handful of SNPs. It is therefore critical to show that the results are robust. While I agree that it is valid to leave the hypothesized contaminant region of BM in the analysis, the authors should also consider an Extended Data analysis omitting this region to show that this does not significantly change the results. The authors already did a similar comparison of the tree phylogenies using ambiguities and majority rule consensus and show that the overall results are sound, but the tree topologies do change somewhat.

Reply: For phylogenetic analyses we had already opted out for a conservative approach and masked the potential recombinant fragment. We have clarified this in the methods, lines 620-621 of the revised manuscript, and annotated it as response to review comment #3.

4. Secondly, the authors should include the analysis of the data excluding the new genomes for the HSLCstd and HSLCext models and dN/dS values in the supplementary information. I disagree that this is unimportant information -- it raises the critical issue of model misspecification in terms of biological/phylogenetic inference. The outcome of Worobey et al. 2014 changes significantly if the model is incorrect.

Reply: We have now included these results in the supplementary information file, at the end of the subsection "HSLC standard versus HSLC extended", and annotated it as response to review comment #4. We are aware that supplementary information should not include figures, but since we have already reached the limit of 10 Extended Data files we have (for now) added this last figure in the supplementary information. Shall the editor request it, we would be happy to move it to an additional Extended Data figure.

5. Fig 3. Light grey and light blue are difficult to read. Please increase contrast with white background or change color palette.

Reply: We have adjusted Figure 3 accordingly by changing light colors to darker tones.

6. Fig 4. I had some difficulty interpreting the meaning of panels A and B at first glance. The figure clarity would be improved by adding a line to the caption explaining that pandemic HA clusters with swine HA in HSLCstd, but with seasonal human HA in HSLCext. Also, the pandemic and seasonal HA use the same symbol. Maybe change the color for the pandemic HA?

Reply: We have clarified the information of the figure by adding "seasonal" and "1918 pdm" next to the respective human silhouettes. We hope this will help readers identify the two sets of sequence more easily.

7. I congratulate the authors on their excellent contribution and I look forward to seeing it published in final form.

Reply: We are grateful to the reviewer for such a kind comment and encouragement.

Reviewer comments, second review -

Reviewer #3 (Remarks to the Author):

Patrono and co-authors have addressed all of my comments. I look forward to seeing their contribution in print.

Michael Campana